# Detecting Brain Anomalies in Clinical Routine with the $\beta$-VAE: Feasibility Study on Age-Related White Matter Hyperintensities

**Sophie Loizillon**[1]             SOPHIE.LOIZILLON@ICM-INSTITUTE.ORG

**Yannick Jacob**[3]              YANNICK.JACOB@APHP.FR

**Aurelien Maire**[3]            AURELIEN.MAIRE@APHP.FR

**Didier Dormont**[1,2]          DIDIER.DORMONT@ICM-INSTITUTE.ORG

**Olivier Colliot**[1]             OLIVIER.COLLIOT@CNRS.FR

**Ninon Burgos**[1]              NINON.BURGOS@CNRS.FR

**APPRIMAGE Study Group**[4]

[1] *Sorbonne Université, Institut du Cerveau - Paris Brain Institute - ICM, CNRS, Inria, Inserm, AP-HP, Hôpital de la Pitié Salpêtrière, Paris, France.*

[2] *AP-HP, Pitié Salpêtrière, DMU DIAMENT, Dep. of Neuroradiology, Paris, France.*

[3] *AP-HP, Innovation & Données – Département des Services Numériques, Paris, France*

[4] *Members of the APPRIMAGE study group can be found at www.aramislab.fr/apprimage*

**Editors:** Accepted for publication at MIDL 2024

## Abstract

This experimental study assesses the ability of variational autoencoders (VAEs) to perform anomaly detection in clinical routine, in particular the detection of age-related white matter lesions in brain MRIs acquired at different hospitals and gathered in a clinical data warehouse (CDW). We pre-trained a state-of-the-art $\beta$-VAE on a healthy cohort of over 10,000 FLAIR MR images from the UK Biobank to learn the distribution of healthy brains. The model was then fine-tuned on a cohort of nearly 700 healthy FLAIR images coming from a CDW. We first ensured the good performance of our pre-trained model compared with the state-of-the-art using a widely used public dataset (MSSEG). We then validated it on our target task, age-related WMH detection, on ADNI3 and on a curated clinical dataset from a single-site neuroradiology department, for which we had manually delineated lesion masks. Next, we applied the fine-tuned $\beta$-VAE for anomaly detection in a CDW characterised by an exceptional heterogeneity in terms of hospitals, scanners and image quality. We found a correlation between the Fazekas scores extracted from the radiology reports and the volumes of the lesions detected by our model, providing a first insight into the performance of VAEs in a clinical setting. We also observed that our model was robust to image quality, which strongly varies in the CDW. However, despite these encouraging results, such approach is not ready for an application in clinical routine yet due to occasional failures in detecting certain lesions, primarily attributed to the poor quality of the images reconstructed by the VAE.

**Keywords:** Anomaly Detection, White Matter Hyperintensities, Clinical Data Warehouse, MRI

## 1. Introduction

Age-related white matter hyperintensities (WMHs) are lesions in the white matter that appear hyperintense on FLAIR MRI and are a very common finding in elderly patients (Debette and Markus, 2010; Prins and Scheltens, 2015; Habes et al., 2016; Wardlaw et al., 2015). Detection of these lesions is a clinically relevant task to assess their severity. In clinical routine, such WMHs are visually evaluated using the Fazekas score, which is a 3-grade scale where 0 corresponds to no lesion and 3 to large, extensive and confluent lesions (Fazekas et al., 1987). One of the main problems with visual rating scales is that they suffer from intra- and inter-subject variability, leading to inconsistencies between studies (Caligiuri et al., 2015). This is why the development of an automatic tool capable of performing WMH detection, robust to MRIs acquired with different machines, manufacturers or acquisition parameters at different sites and of various image quality, is desirable in clinical routine.

In recent years, unsupervised machine learning algorithms for medical imaging have emerged, with the advantage of not requiring costly and time-consuming manual annotation (Chen and Konukoglu, 2022). These algorithms are capable of tackling complex tasks such as anomaly detection (Fernando et al., 2021). Unsupervised anomaly detection (UAD) can be based on generative models such as variational autoencoders (VAEs) (Baur et al., 2021), generative adversarial networks (GANs) (Schlegl et al., 2019), or diffusion models (Wolleb et al., 2022). Trained on healthy brain MRIs, the model learns the distribution of healthy brain tissue, ensuring that when confronted with an image presenting anomalies, the abnormal tissues are not well reconstructed. Comparing the reconstructed image with the real one then allows the detection of anomalies.

Among the deep generative models that have been used for UAD, VAEs have demonstrated their ability to detect lesions in datasets of brain MRIs with diverse pathology including multiple sclerosis (Commowick et al., 2018), glioblastoma (Menze et al., 2014) and cerebral small vessel disease (Kuijf et al., 2019). However, the effectiveness of VAEs on routine clinical datasets that reflect the reality of current practice remains to be demonstrated. Clinical data warehouses (CDWs), which gather medical images from thousands to millions of patients, are representing an exceptional opportunity to perform this kind of validation on images acquired on different machines, with non-standardised acquisition parameters and variable image quality (Bottani et al., 2023; Loizillon et al., 2024).

We propose an experimental study to assess the potential of VAEs, using the state-of-the-art $\beta$-VAE (Higgins et al., 2016), for anomaly detection, and more specifically targeting age-related WMHs, on clinical datasets. After pre-training our VAE on a non-lesional cohort of FLAIR MRIs from the UK Biobank, we will fine-tune it on a unique FLAIR dataset extracted from the Parisian CDW gathering images from up to 39 different hospitals. After validating the effectiveness of our anomaly detection model on research datasets and on a curated clinical dataset, which has been acquired at a single neuroradiology department, we will evaluate the feasibility of its application on routine clinical MRIs from the Parisian CDW known for its exceptional heterogeneity.

## 2. Materials

### 2.1. Construction of the Non-Lesional Cohorts

Research Dataset The UK Biobank (UKB) is a prospective cohort study involving 500,000 participants aged between 40 and 69 years at time of recruitment (2006–2010). (Sudlow et al., 2015). Neuroimages, all acquired on the same type of scanner (Siemens Skyra 3 T) using the same acquisition parameters at different sites, are available for some of the participants (Alfaro-Almagro et al., 2018). In our study, we were only interested in 3D FLAIR MRIs of healthy appearance. After linking the FLAIR images to all diagnostic codes (ICD-10) associated with the patients at each visit, we excluded images in which patients were diagnosed with dementia or lesions. In this way, we extracted a supposedly healthy cohort of 11,990 FLAIR images.

Clinical Data Warehouse We built a healthy cohort with clinical routine data coming from a large CDW containing all the FLAIR brain MRIs of adult patients scanned in hospitals of the Greater Paris area (Assistance Publique-Hôpitaux de Paris [AP-HP]). Within the CDW we had access to 13,703 FLAIR MRIs and developed the following approach to build a non-lesional cohort. We first associated each MRI with its radiological report and any ICD-10 diagnostic codes associated with the patient. This allowed us to perform an initial filter, eliminating all patients with an ICD-10 code related to dementia or the presence of brain lesions. We then analysed the radiological reports using the EDS-NLP tool (Wajsburt et al.), which allowed us to extract the *"Conclusion"* section of the document. A filtering process was applied to these conclusions, looking for references to the absence of abnormalities: *"pas d'anomalie", "normal", "absence d'anomalie", "pas d'argument", "sans anomalie", "pas de signe", "pas d'accident", "sans particularité", "Pas de lésion cérébrale" et "Pas de lésion encéphalique"* (Figure A1). The dataset was further refined by a manual filtering process on the *Conclusion* associated with a visual inspection of the images to ensure that the resulting MRIs correspond to 3D non-lesional FLAIRs (i.e., no straight reject, see below). Thus, we built a new healthy cohort of 674 FLAIR images out of the 13,703 which were acquired on 12 different machines from three different manufacturers. This dataset is characterised by its great heterogeneity with images acquired over 17 different hospitals with no homogenisation in the acquisition parameters. Thus, this cohort well represents 3D FLAIR brain MRIs that may be acquired in other hospitals every day.

### 2.2. Datasets with Images Presenting Lesions

Research Datasets The MSSEG MICCAI challenge, which includes 53 patients affected by multiple sclerosis across four different sites, aims to perform the segmentation of WMHs (Commowick et al., 2018). Four different scanners were used: GE Discovery 3 T, Philips Ingenia 3 T, Siemens Aera 1.5 T and Siemens Verio 3 T. Each patient underwent four MRI sequences: 3D FLAIR, 3D T1w, 3D contrast-enhanced T1w and 2D T2w. In our study, we only considered the 3D FLAIR. The Alzheimer's Disease Neuroimaging Initiative (ADNI) is a multi-site study of elderly individuals with normal cognition, mild cognitive impairment, or Alzheimer's disease (Weiner et al., 2017). The ADNI-3 phase includes 3D FLAIR MRIs acquired exclusively on 3 T scanners from different manufacturers (GE,

Siemens, and Philips). We used 20 of the FLAIR images that had previously been manually segmented by a trained radiology resident as described in (Vanderbecq et al., 2020).

CURATED SINGLE-SITE CLINICAL DATASET This routine clinical dataset consists of 60 patients diagnosed with cognitive impairment at the neuroradiology department of the Pitié-Salpêtrière hospital. We reused the dataset from a previous study (Vanderbecq et al., 2020), in which all patients had 3D T1-w and FLAIR sequences (except for two patients who had a 2D FLAIR). All data were collected during a routine clinical workup and were retrospectively extracted for the purpose of this study. Therefore, according to French legislation, explicit consent was waived. The images were acquired on four different MRI scanners and curated for image quality. Manual segmentation of WMHs was performed by a trained radiology resident (Vanderbecq et al., 2020). This dataset will be referred to as PITIE in the remainder of the article.

CLINICAL DATA WAREHOUSE We constructed a cohort of patients presenting brain lesions using routine clinical data from the AP-HP CDW. We developed the following approach to construct a new cohort out of the 13,703 FLAIR MRIs available. We first linked each FLAIR with its radiological report. From the radiological report, we extracted the Fazekas scale (Fazekas et al., 1987) when it was mentioned in the *"Conclusion"* using EDS-NLP. We found this information for 204 FLAIR images. In contrast to MRIs from the PITIE dataset, these images come from nine different types of machines in 14 hospitals and have a wide range of image quality, reflecting the diversity of MRIs seen in clinical routine. We automatically assessed the image quality using our quality control model, which classifies images into good, medium, low quality and straight reject (e.g., truncated images) (Loizillon et al., 2023). The quality distributions across Fazekas scores are displayed in the appendix (Table A1).

Participant demographics for the various cohorts are summarised in Table 1.

Table 1: Age (average [range]) and sex (% females) of the participants from the non-lesional (UKB & CDW) and lesional (MSSEG, ADNI3, PITIE and CDW) cohorts. Note that for the CDW cohort with images presenting lesions, we had access to demographic data for only 171 out of the 204 images.

|  | Dataset | N images | Age | Sex (%F) |
|---|---|---|---|---|
| Non-lesional cohorts | UKB | 11990 | 56.63 [44-83] | 62.73 |
|  | CDW | 674 | 45.57 [18-87] | 55.49 |
| Cohort with images presenting lesions | MSSEG | 53 | 45.42 [24-66] | 71.70 |
|  | ADNI | 20 | 71.07 [58-83] | 50 |
|  | PITIE | 60 | 78.20 [52–101] | 50 |
|  | CDW | 204 | 77.34 [43-96] | 52.05 |

## 3. Proposed Approach

### 3.1. Image Pre-processing

FLAIR MRIs were pre-processed using the `flair-linear` pipeline from Clinica (Routier et al., 2021). First, a bias field correction was applied using the N4ITK method (Tustison

et al., 2010). An affine registration to the MNI space was then performed (Avants et al., 2008). Registered images were normalised by clipping the intensity values to the [2,98] percentiles and cropped to remove background resulting in images of size $169 \times 208 \times 179$, with 1 mm isotropic voxels.

### 3.2. Unsupervised Anomaly Detection with a $\beta$-VAE

Many VAE variants have been proposed to perform UAD (Chadebec et al., 2022) and several have recently been compared in a benchmark focused on detecting dementia-related lesions in 3D positron emission tomography images (Hassanaly et al., 2023, 2024). Based on the results of this benchmark (Hassanaly et al., 2024), we decided to train a $\beta$-VAE (Higgins et al., 2016), which encourages the disentanglement of features in the latent space by adding a weight $\beta$ in front of the Kullback-Leibler divergence regularisation term to adjust the balance with the reconstruction loss.

We used a 3D $\beta$-VAE model with an encoder of five blocks and a symmetric decoder (Hassanaly et al., 2024), see Figure 1. Each encoder block is composed of a convolutional layer, a batch normalisation and a swish activation function. These blocks are followed by a flatten and a fully connected layer. The latent space size was 256 and $\beta$ equal to 10. The model was trained over 30 epochs, with a learning rate of $10^{-5}$ and a batch size of 4 using ClinicaDL (Thibeau-Sutre et al., 2022).

Figure 1: Variational autoencoder (VAE) architecture for brain anomaly detection

### 3.3. Post-processing

After multiplying each residual image, i.e., the difference between the input and output of the $\beta$-VAE, with an eroded brain mask to eliminate false positives near the brain contour, a 3D median filter with a kernel size of 5 was used to obtain a smoother mask that was thresholded to obtain a binary segmentation mask. As in (Baur et al., 2021), the threshold was model specific and determined as the 98th percentile of the model reconstruction errors on the training dataset. Finally, we performed a 3D connected component analysis by excluding any segmented regions with an area of less than 10 voxels.

### 3.4. Experimental Setup

For pre-training our $\beta$-VAE model on the UKB dataset, the 11,990 images were split into a training and a validation set containing 8304 and 3686 MRIs. In the following fine-tuning step on the CDW, 574 images were used for training and 100 were left for validation. The separation between training and validation sets was done at the subject level to avoid data leakage and stratified by sex and age. We evaluate the pre-trained model on three independent test sets presenting brain anomalies: MSSEG, ADNI3 and PITIE, and apply the fine-tuned model on the CDW.

### 3.5. Evaluation Metrics

To evaluate the anomaly detection results, the two best suited metrics according to (Maier-Hein et al., 2022) were the Dice Similarity Coefficient (DSC) and the Normalised Surface Dice (NSD). We also computed three other relevant metrics: absolute volume error rate (AVR), voxel-level false positive ratio (FPR), voxel-level false negative ratio (FNR).

$$ AVR = \frac{|V_R - V_A|}{V_R} \;\; , \quad FPR = \frac{FP}{FP + TN} \;\; , \quad FNR = \frac{FN}{FN + TP} \;\; , $$

where $V_R$ is the reference volume, $V_A$ is the automatic volume, $FP$ is a false positive voxel, $TN$ is a true negative voxel, $FN$ is a false negative voxel and $TP$ is a true positive voxel. For each metric, we report the mean and the 95% confidence interval computed using bootstrapping on the corresponding independent test set (9999 resamples).

## 4. Results

### 4.1. Validation of the $\beta$-VAE for Anomaly Detection in Research Datasets

We first tested our pre-trained model using the MSSEG dataset, which has been used extensively in the literature to detect multiple sclerosis lesions. Although these lesions differ from age-related WMHs, our target, this step ensures that our VAE produces results consistent with those reported in the literature on a publicly available dataset. We then validated our model on our target task – the anomaly detection of age-related WMHs – on ADNI3 and the curated clinical dataset PITIE. Results are presented in Table 2.

Table 2: Validation on research datasets (MSSEG and ADNI3) and a curated clinical dataset (PITIE). Each metric is presented as average [95% confidence interval].

| Dataset | FLAIR | DSC (%) | NSD (%) | AVR | FPR | FNR |
|---------|-------|---------|---------|-----|-----|-----|
| MSSEG | 53 | 30.81 [25.66,36.07] | 32.05 [26.63,37.34] | 0.80 [0.64,0.98] | 0.63 [0.55,0.71] | 0.62 [0.58,0.66] |
| ADNI3 | 20 | 27.06 [19.73,34.52] | 30.68 [22.98,38.4] | 0.71 [0.61,0.8] | 0.77 [0.68,0.85] | 0.48 [0.4,0.56] |
| PITIE | 60 | 35.02 [29.89,40.2] | 36.63 [31.29,41.81] | 0.64 [0.56,0.71] | 0.71 [0.65,0.77] | 0.4 [0.35,0.44] |

The DSC of 30.81% observed on MSSEG is in agreement with existing works using VAEs for WMH detection on this dataset, such as that of (Baur et al., 2021), which obtained a DSC of 25.70%. We obtained similar results on ADNI3 and PITIE, showing that we were able to detect age-related WMHs. FPR was higher on the ADNI3 and PITIE datasets, which may be attributed to the difficulty of the model to reconstruct areas with atrophy. In contrast, we observe a higher FNR on the MSSEG dataset, suggesting that our model fails to detect many lesions of multiple sclerosis patients, which present different shapes and contrasts distributions compared to age-related WMHs. A sample from the PITIE dataset is presented in Figure 2. Further examples are given in the appendix (Figure A2).

Figure 2: Left to right: input MRI, reconstructed MRI, residual map, post-processed lesion map, ground truth. This example corresponds to a favourable outcome in terms of DSC in regard to the overall results.

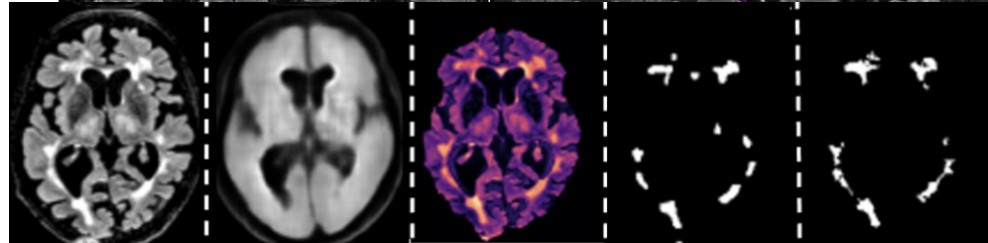

### 4.2. Application of the $\beta$-VAE on Clinical Routine Images

As observed in (Bottani et al., 2023; Loizillon et al., 2024), there is an important drop of performance when applying a model trained on research data to heterogeneous clinical data. This is why we fine-tuned our pre-trained model on 674 healthy FLAIRs from the CDW before applying it on 204 patients of the CDW for which we knew the Fazekas score. Figure 3 depicts the lesion maps obtained for Fazekas scores 1, 2, and 3. More examples are shown in the appendix (Figure A5).

Figure 3: Automatic lesion maps for patients with Fazekas score 1 (A), 2 (B), and 3 (C). A is of medium image quality, B of good image quality and C of low image quality.

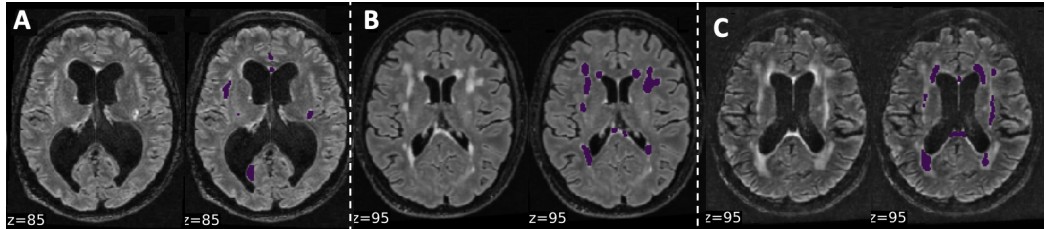

We examined the relationship between the Fazekas score and the volume of lesions automatically obtained by our $\beta$-VAE (Figure 4). Specifically, differences in volumes between the three patient groups with different Fazekas scores (Fazekas 1, Fazekas 2, Fazekas 3)

were assessed using one-way ANOVA. The results were statistically significant (F=32.38, p< $10^{-10}$). Tukey post hoc tests were performed to assess differences between pairs of groups. All pair-wise differences were significant (Fazekas 1 vs 3: $p = 0.0036$, Fazekas 1 vs 2: $p < 0.001$, Fazekas 2 vs 3: $p < 0.001$).

Figure 4: Boxplot depicting the distribution of lesion volumes per Fazekas score (left) and per Fazekas score categorised by image quality (right). Every box is bounded by the lower and upper quartiles, with the centre line representing the median. Whiskers extend from the box to the furthest data point within $1.5\times$ the interquartile range of the box. **: statistically significant for Tukey post-hoc test.

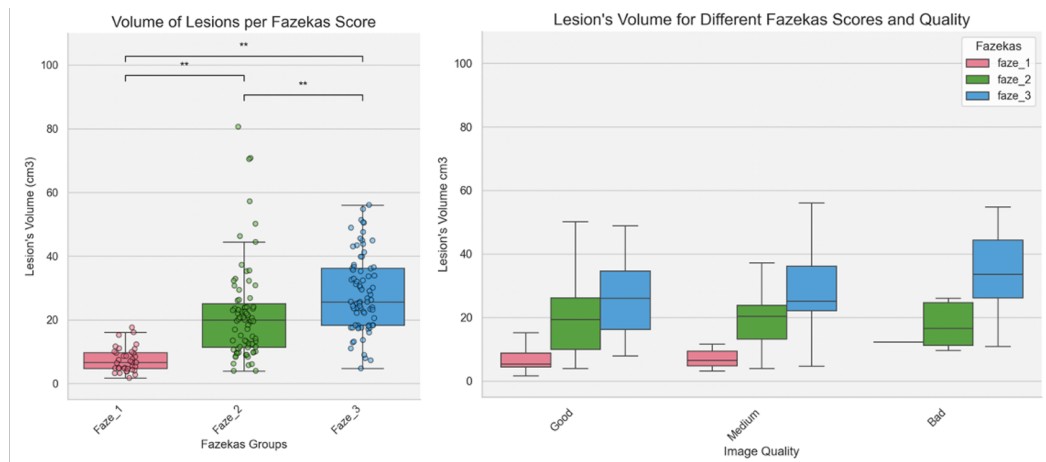

As images from the CDW present a wide range of image quality, we assessed whether our anomaly detection model was robust to quality. In Figure 4, we plot for every quality level the distribution of WMH volumes by Fazekas score. We did not perform statistical testing due to the very small sample size in some subclasses. Nevertheless, the graph qualitatively shows the robustness of our model to image quality since the order in volumes across Fazekas classes is preserved for all quality levels.

## 5. Conclusion

In this feasibility study, we assessed the ability of VAEs to detect age-related WHMs on clinical routine MRIs acquired at different hospitals and gathered in a CDW. Despite, the promising results found – correlation between the Fazekas scores and the volumes of the lesions detected by our model; robustness to image quality – such models are not ready for a clinical routine application yet. This may be due to a systematic failure to reconstruct details of the cortical and subcortical brain structures, resulting in difficulties in identifying some WMHs and leading to failure cases where no WMH is detected at all in Fazekas 2 images (Figure 4). A limitation of our study is the use of a $\beta$-VAE whose architecture and hyperparameters were optimised on a different imaging modality (Hassanaly et al., 2024). It may therefore be of interest to optimise the architecture and hyperparameters for our specific use case. In addition, it would be interesting to evaluate the ability of this model to detect other types of WMHs, such as multiple sclerosis lesions. This is left for future work.

## Acknowledgments

The research leading to these results has received funding from the French government under management of Agence Nationale de la Recherche as part of the "Investissements d'avenir" program, reference ANR-19-P3IA-0001 (PRAIRIE 3IA Institute) and reference ANR-10-IAIHU-06 (Agence Nationale de la Recherche-10-IA Institut Hospitalo-Universitaire-6).

The research was done using the Clinical Data Warehouse of the Greater Paris Hospitals. The authors are grateful to the members of the AP-HP DSN and URC teams, and in particular Stéphane Bréant, Florence Tubach, Jacques Ropers, Pierre Rufat, Antoine Rozès, Camille Nevoret, Christel Daniel, Martin Hilka, Julien Dubiel, Cyrina Saussol and Rafael Gozlan. They would also like to thank the "Collégiale de Radiologie of AP-HP" as well as, more generally, all the radiology departments from AP-HP hospitals.

Data used in preparation of this article were obtained from the Alzheimer's Disease Neuroimaging Initiative (ADNI) database (adni.loni.usc.edu). As such, the investigators within the ADNI contributed to the design and implementation of ADNI and/or provided data but did not participate in analysis or writing of this report. A complete listing of ADNI investigators can be found at: http://adni.loni.usc.edu/wp-content/uploads/how_to_apply/ADNI_Acknowledgement_List.pdf.

The authors would also like to thank Ravi Hassanaly and Maëlys Solal for their help implementing VAEs and their feedback.

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

## Appendix A.  Construction of Non-Lesional Cohorts

Figure A1: Translation of the terms used to construct the non-lesional cohort in the CDW.

| French | English |
|---|---|
| Pas d'anomalie | No anomaly |
| Normal | Normal |
| Absence d'anomalie | Absence of anomaly |
| Pas d'argument | No argument |
| Sans anomalie | No anomaly |
| Pas de signe | No sign |
| Pas d'accident | No accidents |
| Sans particularité | No particularities |
| Pas de lésion cérébrale | No cerebral lesion |
| Pas de lésion encéphalique | No encephalic lesion |

## Appendix B.  Results of the pre-trained $\beta$-VAE

Figure A2: Automatic WMH maps for two patients of MSSEG (left), PITIE (middle) and
ADNI3 (right). Green voxels correspond to true positives, red to false positives
and blue to false negatives.

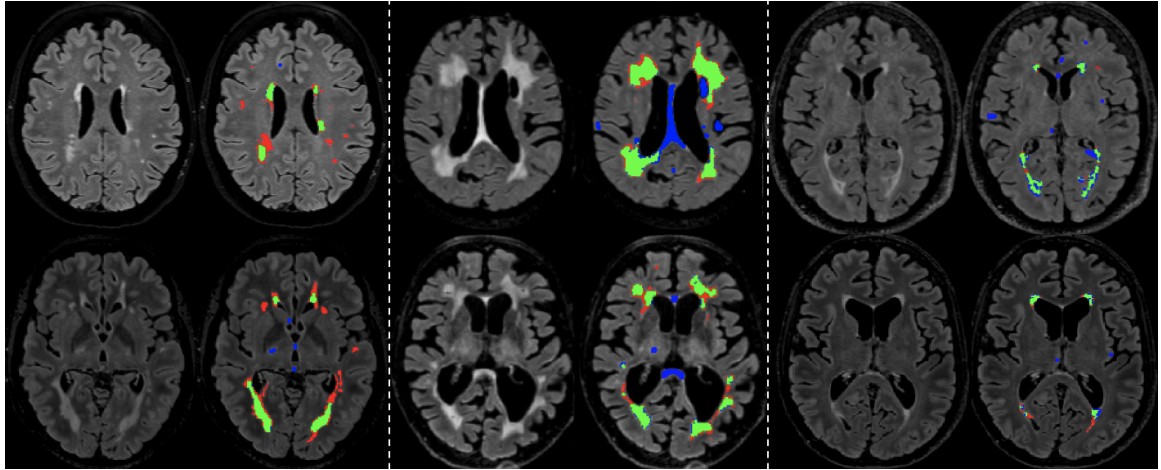

It is important to note that because of the exclusion criteria applied by ADNI, we found
a lower vascular burden in this dataset compared to others (cf. Figure A2).

Figure A3: Automatic lesion maps for cognitively normal patients of ADNI dataset.

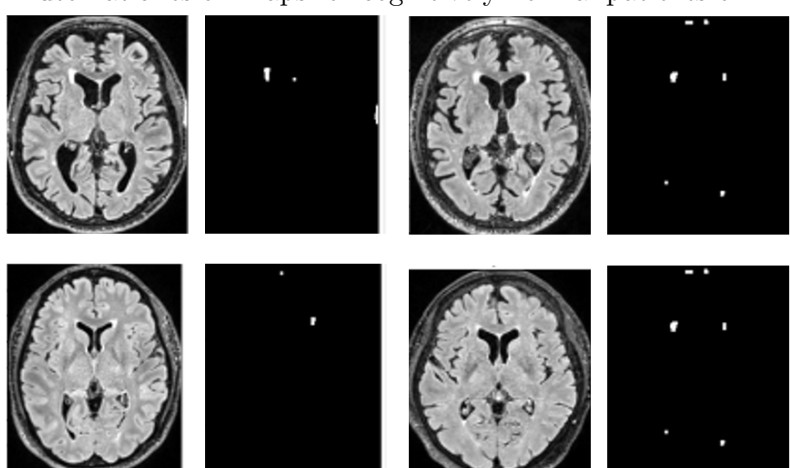

We evaluated our $\beta$-VAE on 25 ADNI subjects labelled as cognitively normal (CN). For each MRI, we computed the amount of lesional volume and obtained a mean of 9.30 cm$^3$ lesional tissues for an average age of 66.87 years. Even if these images are labelled as CN as they are normal related to the age of the patients, they are still presenting some lesions (cf. Figure A3). The validation of our model on a young healthy cohort such as the UKB is left for future work.

In Figure A4, we plotted the lesion burden volume as a function of age for each Fazekas group in order to observe the robustness of our model to age. We visually notice that for the Fazekas 1 group, the volume of lesion remains almost always below 10 cm$^3$ even for patients over 80 years. Same trends were obtained for Fazekas 2 and 3. Thus, we believe our model to be robust to age.

Figure A4: Plot illustrating the variation of lesion volume with age across different Fazekas groups.

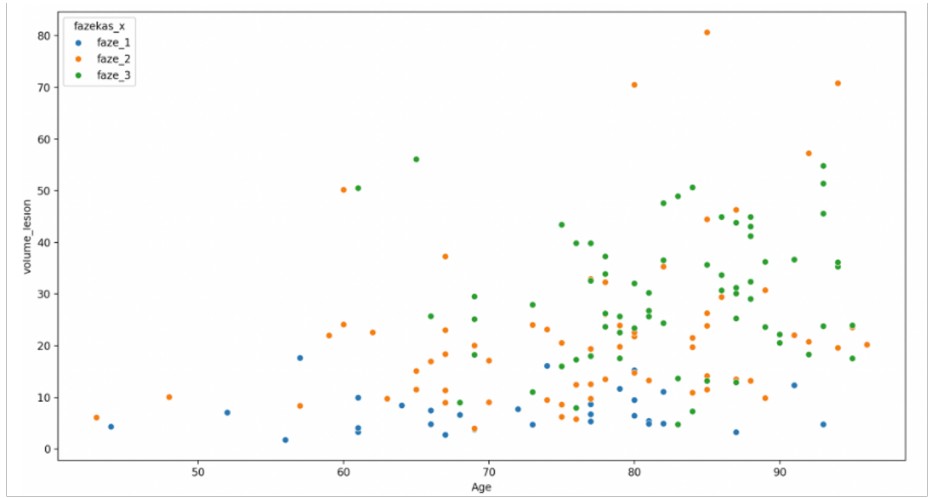

## Appendix C. Quality Levels Across Fazekas Scores

Table A1: Number of images across quality levels and Fazekas scores. Straight reject images are not proper 3D FLAIRs (e.g., images of segmented tissues or truncated images).

|  | Good quality | Medium quality | Low quality | Straight reject |
|---|---|---|---|---|
| Fazekas 1 | 18 | 15 | 1 | 0 |
| Fazekas 2 | 29 | 36 | 6 | 0 |
| Fazekas 3 | 12 | 37 | 14 | 8 |

Table A2: Age (average $\pm$ standard deviation) and sex (% females) across Fazekas scores

|  | Age | Sex (% F) |
|---|---|---|
| Fazekas 1 | $71.17 \pm 11.06$ | 57.14 |
| Fazekas 2 | $75.99 \pm 11.81$ | 39.44 |
| Fazekas 3 | $82.2 \pm 7.78$ | 60.87 |

Figure A5: Automatic WMH maps for two patients with Fazekas score 1 (left), 2 (middle), and 3 (right). A, C and D are good quality MRIs, B and E medium quality images and F a low quality image.

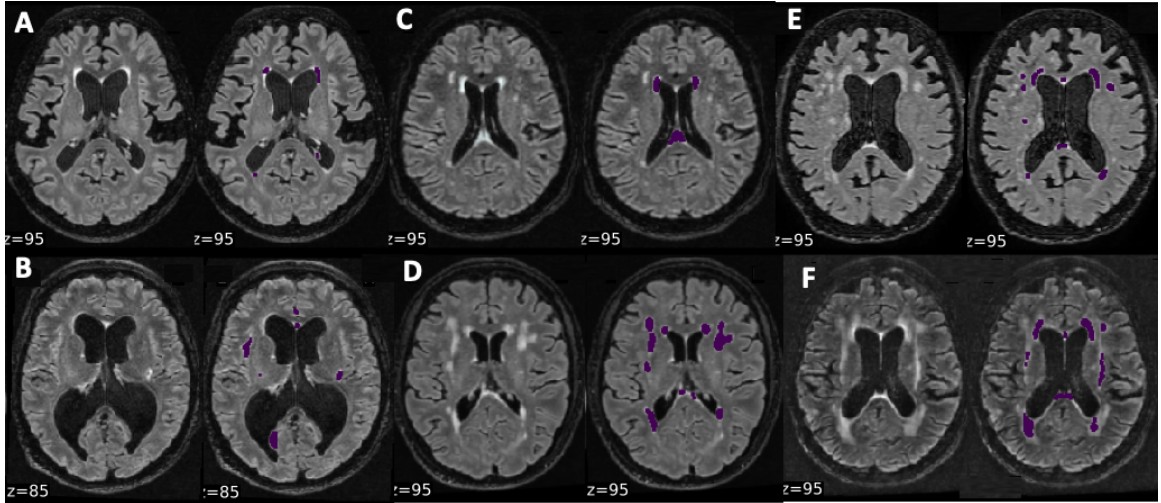

