# OpenReview forum: "Detecting Brain Anomalies in Clinical Routine with the $\beta$-VAE: Feasibility Study on Age-Related White Matter Hyperintensities"
_MIDL.io/2024/Conference — MIDL 2024 Oral_

### Official Review · Reviewer_c8Uz · 2024-02-24

**Confidence:** 4
**Preliminary Rating:** 5
**Recommendation:** Oral
**Final Rating:** 5

**Summary:**

The paper is experimental. It proposes to evaluate the efficacy of detecting brain anomalies (i.e., WMHs) using a beta-VAE trained on non-lesional cohorts and tested on public databases and clinical data warehouse from a cohort with subjects with lesions.

**Strengths:**

The paper is well-written, and it presents an extensive evaluation on public and private data. The results are compelling, especially the fact that the regions detected by the beta-VAE are predictive of the FAzekas score.

**Weaknesses:**

In addition to the limitations stated by the authors in the paper, I believe that Figure 2 is not representative of the Dice score reported in Table 2.  Also, it is unclear why the authors refer to Figure 3 as "lesion maps". Isn't Figure 3 showing the original image and the one reconstructed by the beta-VAE?

**Detailed Comments:**

The paper is well-written and presents a comprehensive evaluation of the results using a combination of public and private data. I suggest improving Figure 2 and adjusting the caption for Figure 3.

**Justification Of Final Rating:**

I think this is an interesting validation paper, and I am supportive of its publication at 2024 MIDL. The amount of data used is quite impressive, as well as the predictive power of the model in determining the Fazekas scores.

**Justification Of The Preliminary Rating:**

The paper is a strong experimental evaluation paper that leveraged nearly 13,000 images. It could be used to inform future clinical applicability of unsupervised anomaly detection models in brain imaging.

Nice work by the authors.

**Questions To Address In The Rebuttal:**

N/A

**Special Issue:**

Yes

---

> ### Author Response · Authors · 2024-03-17
> **Official Comment c8Uz**
>
> We thank reviewer c8Uz24 for underlining the strengths of our experimental paper to evaluate the use in a clinical setting of beta-VAE for the anomaly detection of age-related white-matter hyperintensities. As recommended by the reviewer, we clarify Figure 2 and Figure 3 captions.
>
>
> > In addition to the limitations stated by the authors in the paper, I believe that Figure 2 is not representative of the Dice score reported in Table 2.
>
> True, we modified the caption of the figure indicating that the image corresponds to one of the best results of our model on the test set.
>
> > Also, it is unclear why the authors refer to Figure 3 as "lesion maps". Isn't Figure 3 showing the original image and the one reconstructed by the beta-VAE?
>
> We clarified the caption: on the left the original image and on the right the original image with the segmentation maps

---

> > ### Comment · Reviewer_c8Uz · 2024-03-26
> > **Final review comment**
> >
> > Thank you for your response. I think this is an interesting validation paper and I am supportive of it being published at 2024 MIDL.

---

### Official Review · Reviewer_S4iB · 2024-02-29

**Confidence:** 5
**Preliminary Rating:** 4
**Recommendation:** Oral
**Final Rating:** 5

**Summary:**

In this paper, the authors tackle the task of detecting white matter hyperintensities on clinical routine MRI. Public databases are used for model training and preliminary evaluation, and the model is then fine-tuned and evaluated on the (private) large clinical routine MRI. Authors show, after confirmation that the model has very roughly sota performances on public datasets, that the application of VAE reconstruction error for clinical routine WMH detection seems ~feasible, through a proxy task (correlation of detected volume with Fazekas score).

**Strengths:**

1) This papers evaluate the use of a popular simple UAD method (VAE) on a clinical routine dataset. This task, which the reviewer thinks is particularly difficult and time-consuming (retrieving and pre-processing the data), is of great interest for the community, as it builds a step forward to the use of UAD methods in clinical practice. The reviewer believes this strength single-handedly justifies the acceptance of such a paper.
2) The paper is overall well written and the experiment design is clear

**Weaknesses:**

1) The reviewer find some small weaknesses or comments that are mentioned bellow but no strong, unchangeable weakness in this paper.
2) The reviewer has nothing more to say but must reach the minimum character limit.

**Detailed Comments:**

- "Unsupervised anomaly detection (UAD) is based on generative models such as variational autoencoders" : I think this sentence has to be nuanced, UAD employs a wide variety of methods (that can be classified into 1) reconstruction based 2) classification based 3) density estimation based, see the great review of anomaly detection by Ruff et al. 2021), and VAE can serve to do UAD in several ways, not only by their generative properties.
- "Thus, we built a new healthy cohort of 674 FLAIR images which were acquired on 12 different" out of the 13k from the CDW right ?
- In Table 1 (24-66) should be in []
- Table 2 the Dice is in %, it could be written.
- "leading to failure cases where no WMH is detected at all in Fazekas 2 images (Figure 4)" : either this points to the wrong figure or the reviewer didn't understand the point being made.

**Justification Of Final Rating:**

The reviewer was already fairly confident that this work should be presented at the MIDL conference and the authors did a great job at addressing the comments and made a clear response. I highly support this work.

**Justification Of The Preliminary Rating:**

The reviewer would strongly accept this contribution if most of the questions to address are answered but as is the paper is still of interest for the community, especially since its application is on clinical routine datasets (which is often overlooked in the literature).

**Questions To Address In The Rebuttal:**

- For the evaluation metrics, the authors refer to a paper indicating the best practice for choosing metrics. Could the authors precise how this paper help pick the metrics and how ? To be very picky : Furthermore the paper is listed after Dice, as if Dice was coined by this paper.
- "The DSC of 30.81% observed on MSSEG is in agreement with existing works using VAEs for WMH detection on this dataset (Baur et al., 2021)." can the authors please report said-values for comparison ?
- The reviewer might have missed some things but how come is there no Faze 1 of bad quality ? Can there be a causation between bad quality and high Faze score ? e.g. motion blurs caused by tremors of old age patients ?
- "such models are not ready for a clinical routine application yet" could the authors detail this statement ? What in the authors view, would make the models ready for clinical application ? What is precisely missing ?
- "This is mainly due to the suboptimal quality of the reconstructed images by the VAE, resulting in difficulties in identifying some WMHs" : the reviewer disagrees with this statement. Poor reconstruction quality does not imply inability to identify the WMH, as with the right treshold, the reconstruction of the WMH only need to be poorer than the reconstruction of the healthy tissues. Furthermore, this claim is not supported by any figure or results in the paper.
- It has been proven (Meissen 2021 "challenging...") that state-of-the-art UAD methods and especially reconstruction based methods can be outperformed by simple thresholding, as many of the UAD methods are evaluated on hyperintense lesion detections. Could the authors implement a simple thresholding routine and provide the performances as baseline results for the research datasets and clinical routine datasets ? The reviewer believes (could be wrong) this experiment is quite straight forward and not too computationally heavy.
- The reviewer is wondering if the age-gap between the controls and the patient have caused the model to underperform on known age-dependant structures, such as cortex borders (brain shrinkage due to age)

**Special Issue:**

No

---

> ### Author Response · Authors · 2024-03-17
> **Official Comment S4iB**
>
> We would like to thank reviewer S4iB for acknowledging the strengths of our study.
> >“Unsupervised anomaly detection is based on generative models such as variational autoencoders" : I think this sentence has to be nuanced
>
> We thank the reviewer for mentioning this point. We modified the manuscript by nuancing this point (cf. Introduction).
>
> >"Thus, we built a new healthy cohort of 674 FLAIR images which were acquired on 12 different" out of the 13k from the CDW right ?
>
> Yes, the 674 healthy FLAIR of the CDW were identified out of the 13k ones. This has been clarified in the manuscript (cf. 2.1).
>
> >"failure cases where no WMH is detected at all in Fazekas 2 images (Figure 4)" : either this points to the wrong figure or the reviewer didn't understand the point being made.
>
> On Figure 4, we can see thanks to the scatter plot that for Fazekas 2 and Fazekas 3 images, our model fails to detect lesions, which leads to a volume around 0. We clarified this in the new version of the manuscript.
>
> > For the evaluation metrics, the authors refer to a paper indicating the best practice for choosing metrics. Could the authors precise how this paper help pick the metrics and how ?
>
> We used Metrics Reloaded (MR) and in particular its online tool to select the best suited metrics for our anomaly detection task. According to it, the Dice and the Normalised Surface Dice were the most meaningful ones. Thus, it is not that Dice is coined by MR but that the selection process of MR led to these two metrics. These two metrics as well as the absolute volume error rate (AVR), voxel-level false positive ratio (FPR), voxel-level false negative ratio (FNR) were reported to perform our quantitative analysis. The sentence has been modified in the new version of the manuscript (cf. 3.5 Evaluation Metrics).
>
> > "The DSC of 30.81% is in agreement with existing works" can the authors please report said-values ?
>
> The VAE and context-VAE implemented by Baur and tested on MSSEG obtained respectively a dice score of 25.70% and 33.60%.
>
> >The reviewer might have missed some things but how come is there no Faze1 of bad quality ? Can there be a causation between bad quality and high Faze score?
>
> In Table A1 of the appendix we reported the number of images across quality levels and Fazekas scores. Only one MRI of Fazekas score 1 has been graded as bad quality by our automatic quality control model. This QC model was trained with a large variety of images of healthy and unhealthy patients with various diseases and has been proven to be robust to age.
>
> >"such models are not ready for a clinical routine application yet" could the authors detail this statement ? What in the authors view?
>
> We believe that such models are not ready for a clinical routine application yet because of the improvable quantitative results and especially due to the remaining failure cases in image rated as Fazekas 2 or 3, where no lesions are detected.
>
> > "This is mainly due to the suboptimal quality of the reconstructed images by the VAE, resulting in difficulties in identifying some WMHs" : the reviewer disagrees with this statement. Poor reconstruction quality does not imply inability to identify the WMH, as with the right treshold, the reconstruction of the WMH only need to be poorer than the reconstruction of the healthy tissues.
>
> We agree on the fact that the reconstruction of the WMH only needs to be poorer than the reconstruction of the healthy tissues. Nevertheless, we observed a systematic failure to reconstruct details of the cortical and subcortical brain structures. We nuanced this statement in the conclusion.
>
> > It has been proven (Meissen 2021 "challenging...") that state-of-the-art UAD methods and especially reconstruction based methods can be outperformed by simple thresholding, as many of the UAD methods are evaluated on hyperintense lesion detections. Could the authors implement a simple thresholding routine and provide the performances as baseline results for the research datasets and clinical routine datasets ?
>
> We implemented a simple threshold method to compare our approach with a simple baseline. We obtained the following Dice results :
> - ADNI : 8.25%
> - MSSEG : 12.93%
> - CS : 14.39%
>
> A proper comparison with a histogram equalisation followed by a threshold optimization per dataset as proposed by (Meissen et al. 2021), which we believe will greatly improve the results of the proposed baseline, is left for future work.
>
> > The reviewer is wondering if the age-gap between the controls and the patient have caused the model to underperform on known age-dependant structures
>
> We added in the appendix a plot of volume lesion according to the age for the different Fazekas groups (cf. Figure A4 in the appendix). We visually notice that for the Fazekas 1 group, the volume of lesion remains almost always below 10 cm3 even for patients over 80. Same trends were obtained for Fazekas 2 and 3. Thus, we believe our model to be robust to age.

---

> > ### Comment · Reviewer_S4iB · 2024-03-26
> >
> > The reviewer thanks the authors for the additional comments and studies made. This has made the manuscript even clearer and relevant for the community.
> > One minor comment still: "The VAE and context-VAE implemented by Baur and tested on MSSEG obtained respectively a dice score of 25.70% and 33.60%" could the authors add this sentence in the manuscript ? The reviewer believes it is a small addition which is of great confort for the reader of the paper and that does not take too much space.

---

> > > ### Author Response · Authors · 2024-03-28
> > > **Official Comment S4iB**
> > >
> > > We thank the reviewer for his comments. We address his last minor comment by adding the dice score obtained by Baur to the manuscript.

---

> > > > ### Comment · Reviewer_S4iB · 2024-03-28
> > > >
> > > > Thanks again for this nice work

---

### Official Review · Reviewer_BTi2 · 2024-02-29

**Confidence:** 5
**Preliminary Rating:** 3
**Recommendation:** Poster
**Final Rating:** 3.5

**Summary:**

In this paper, the authors propose a feasibility study for the use of B-VAE for the segmentation of age-related white-matter hyperintensities when using clinical grade scans.
They train their system on what they consider healthy research graded scans from UK Biobank and from a Clinical Database Warehouse and apply it to different datasets (both research graded and clinical scans)
They propose an evaluation based on segmentation comparison and on correlation with visual scales. They further assess the impact of image quality on their results

**Strengths:**

- Well justified setting of the problem regarding both lack of annotation, variability of clinical scans and differences between research and clinical settings
- Solid statistical representation and justification of the results
- Clear setting of methods and definition of healthy and unhealthy datasets
- Balanced conclusion in adequation with the results and acknowledgment of limitations

**Weaknesses:**

- The definition of healthy scans for UKB is based on the disease classification score while it could be based on the extracted information regarding WMH present in the database. Often there may be no indication of lesion in the report if it is deemed normal for age but that may not mean that those are truly absent
- The evaluation does not include any evaluation of distance metrics (Hausdorff distance or Normalised Surface Dice) which would be particularly relevant.
- They use the MSSEG dataset which is not a dataset related to WMH but to MS lesions which present slightly differently and in a different population (while the focus of the paper is strongly put on age-related WMH).
- There is some lack of clarity regarding the caption of certain figures (what are the tiers, which one correspond to best quality in Figure 4
- It seems that the system is only evaluated on lesion data but it would be important to assess whether anything is reported on deemed healthy scans.

**Detailed Comments:**

In addition to the main points mentioned above, it would be good to clarify some of the captions:
- what does quality refer to in Figure 3 - quality of the segmentation or quality of the scan?
- why not using the WMH segmentation challenge from 2017?
- A comment on the natural low vascular load of ADNI3 due to exclusion criteria may be warranted.
- Given the strong relationship between DSC and volume, an indication of the range of expected volumes for the different tested datasets would be necessary.
-

**Justification Of Final Rating:**

The paper is interesting and the authors made a real effort in addressing the comments. It is probably worth discussing around a poster and there may be some interesting avenues to consider for an extension with different definition of healthy for a journal paper

**Justification Of The Preliminary Rating:**

The paper is overall quite sound but there are limitations regarding the definitions, the chosen datasets and the methods of evaluation. A stronger emphasis on the true novelty may have helped in highlighting the added value of the work which can be easily lost.

**Questions To Address In The Rebuttal:**

Please do address main points regarding definition of healthy examples and of testing in healthy cases.

**Special Issue:**

No

---

> ### Author Response · Authors · 2024-03-17
> **Official Comment BTi2**
>
> We thank reviewer BTi2 for underlining the strengths of our study on the use of beta-VAE in clinical routine for the anomaly detection of age-related white-matter hyperintensities.
>
> > The definition of healthy scans for UKB is based on the disease classification score while it could be based on the extracted information regarding WMH present in the database. Often there may be no indication of lesion in the report if it is deemed normal for age but that may not mean that those are truly absent.
> >
>
> We thank the reviewer for mentioning this point. We extracted the total white matter lesion volume for our cohort of healthy UKB subjects and obtained a mean white matter lesion volume of 4,115cm3 +/- 5,412cm3.
>
> We calculated the mean volume by age category:
>
> - For subjects under 50: 1.867cm3
> - For subjects aged between 50 and 60: 2.281cm3
> - Subjects aged between 60 and 70: 4.380cm3
> - For subjects aged between 70 and 80: 7.690cm3
>
> It is important to note that white matter lesions are very common with a prevalence of over 90% in the over 65 age group (Wharton et al. 2015). Thus, despite the presence of small volumes of lesions, we considered these brains to be normal for their age. A more restrictive filter directly based on WMH volume would have limited our healthy cohort to younger patients and would not have been representative of the overall healthy population.
>
> >The evaluation does not include any evaluation of distance metrics (Hausdorff distance or Normalised Surface Dice) which would be particularly relevant.
>
> We have calculated the Normalised Surface Dice (NSD) for the three datasets in order to add a distance metric to the evaluation (cf. Table 2):
> - MSSEG : 32.05%
> - ADNI : 30.68%
> - CS : 36.63%
>
>
> >They use the MSSEG dataset which is not a dataset related to WMH but to MS lesions which present slightly differently and in a different population (while the focus of the paper is strongly put on age-related WMH).
>
> Indeed the focus of this article is the detection of age-related WMH. We agree with the reviewer that MS lesions different from WMH in several aspects (appearance, location, affected populations). The purpose of using MSSEG was to validate our model on a publicly available dataset that is widely used for unsupervised anomaly detection. By doing so, we were able to compare our results with the existing literature before drawing any conclusions about the use of such models in clinical routine (cf. 4.1 Validation of the beta-VAE for Anomaly Detection in Research Datasets).
>
>  >There is some lack of clarity regarding the caption of certain figures (what are the tiers, which one correspond to best quality in Figure 4.
>
> We apologise for this lack of clarity. Tiers 1, 2 and 3 correspond respectively to images of good, average and poor quality, referring to our previous work (Bottani 2022; Loizillon 2023). In the new version of the manuscript, we have modified Figure 4 so as not to mention the tiers
>
>
> > It seems that the system is only evaluated on lesion data but it would be important to assess whether anything is reported on deemed healthy scans.
>
> We evaluated our beta-VAE on 25 ADNI subjects labelled as cognitively normal (CN). For each MRI, we compute the amount of lesional volume and obtain a mean of 9.30cm3 lesional tissues for an average age of 66.87 years. Even if these images are labelled as CN as they are normal related to the age of the patients, they are still presenting some lesions (cf. Figure A3 in the appendix). It would be interesting to validate our model on a healthy young cohort such as the UKB. This is left for future work.
>
> > What does quality refer to in Figure 3 - quality of the segmentation or quality of the scan?
>
> In Figure 3, the quality of the scan refers to the quality of the image. This has been  clarified in the caption of Figure 3.
>
> > Why not using the WMH segmentation challenge from 2017?
>
> We did not use the WMH segmentation challenge because we developed a 3D anomaly detection approach. Indeed UMC Utrecht and NUHS Singapore sites only acquired 2D FLAIR images and in the other site where a 3D FLAIR was acquired, it was reoriented transversally and resampled to a slice-thickness of 3.00 mm making it hard to use in the scope of our pre-processing workflow on the clinical data warehouse (cf. Section 3.1 Pre-Processing). https://wmh.isi.uu.nl/#\_Toc122355654
>
> > A comment on the natural low vascular load of ADNI3 due to exclusion criteria may be warranted.
>
> We added in the updated manuscript (cf. Appendix B) a sentence mentioning the low vascular burden in ADNI due to the exclusion criteria.
>
>
> > Given the strong relationship between DSC and volume, an indication of the range of expected volumes for the different tested datasets would be necessary.
>
> We agree that there is a strong link between the Dice score and the volume. Would you like us to compute the average volume of lesion burden for each dataset and then compare it with the Dice score obtained?

---

### Author Response · Authors · 2024-03-17
**Official Comment**

We thank the reviewers for underlining the strengths of our feasibility study for the use of beta-VAE for the anomaly detection of age-related white-matter hyperintensities in clinical routine. As mentioned by R2, we believe our work would be of great interest for the community, building a step forward to the use of unsupervised anomaly detection methods in clinical routine. Our methodology was described as clear by the three reviewers and the solid statistical representation and justification of the results were also underlined.


We addressed during the rebuttal phase the following points raised by the reviewers:

- We tested our anomaly detection model on a new test set of 25 cognitively normal participants of ADNI to evaluate its behaviour on normal brain images for the age of the subjects.
- We extracted the white matter hyperintensities volume for our UKB non lesional cohort to confirm that these subjects' brain images were normal for their age.
- We added a new distance-based metric (Normalised Surface Dice) to complete our quantitative results as suggested by R1.
- We compare our approach with a basic thresholding of the FLAIR, following the same strategy as  (Meissen 2021) recommended by R2.

---

### Meta-Review · Area_Chair_sJdZ · 2024-04-02

**Recommendation:** Accept (Oral)
**Confidence:** 5

**Metareview:**

This work corresponds to an experimental study examining the ability of state-of-the art technique to detect brain abnormalities in routine clinical practice. The standard B-VAE architecture and a simple post-processing strategy were used, and several experiments were carried out to draw clear conclusions.

The strengths of this work are:
1) The training of the model on a large scale dataset 11 990 3D FLAIR volumes and a fine-tuning on a clinical data warehouse dataset composed of 674 FLAIR volumes acquired over 17 different hospitals with no heterogeneous settings.
2) A solid experimental part with clear messages

The main weakness of this work is:
1) The lack of methodological novelty. For example, more advanced techniques for post-processing (detection of region with lesion from the residual map) could have been explored in this article.

This study provides new insights into the automatic detection of brain anomalies in clinical context. The authors have done an excellent job of answering the reviewers' questions, which improves the quality of their paper.

For all these reasons, I have decided to accept this article.

---

### Decision · Program_Chairs · 2024-04-06

Accept (Oral)